# Kinetics of IgG Antibodies in Previous Cases of Dengue Fever—A Longitudinal Serological Survey

**DOI:** 10.3390/ijerph17186580

**Published:** 2020-09-09

**Authors:** Qilin Wu, Qinlong Jing, Xiujuan Wang, Lili Yang, Yilan Li, Zongqiu Chen, Mengmeng Ma, Zhicong Yang

**Affiliations:** 1School of Public Health, Guangdong Pharmaceutical University, Guangzhou 510310, China; gzwuqilin@163.com; 2Guangzhou Center for Disease Control and Prevention, Guangzhou 510440, China; gzcdc_jingql@gz.gov.cn (Q.J.); gzcdc_liyilan@gz.gov.cn (Y.L.); gzcdc_zhaozy@gz.gov.cn (Z.C.); gzcdc_mamm@gz.gov.cn (M.M.); 3School of Public Health, Sun Yat-sen University, Guangzhou 510080, China; wangxj228@mail2.sysu.edu.cn; 4Liwan District Center for Disease Control and Prevention, Guangzhou 510176, China; lwcdcmf@gz.gov.cn

**Keywords:** dengue fever, antibody, kinetics, longitudinal investigation

## Abstract

Guangzhou is believed to be the most important epicenter of dengue outbreaks in southern China. In this study, a longitudinal serological investigation of previous cases of dengue fever in Guangzhou was conducted to explore the persistence of IgG antibodies and related factors affecting the changes of antibody level. We recruited 70 dengue virus type 1 (DENV-1) primary infection cases at two years post infection for serological investigation and conducted a second follow-up in the 5th year of prognosis. An enzyme-linked immunosorbent assay (ELISA) for DENV IgG antibody was examined in all study subjects. Potential factors associated with the concentration of serum total IgG antibody were determined by the generalized estimation equation (GEE). No significant difference in serum total IgG antibody positive rate between two follow-ups was observed (*χ*^2^ = 3.066, *p* = 0.080). However, there was a significant difference in the concentration of serum total IgG antibody between the two follow-ups (Z = 7.154, *p* < 0.001). The GEE showed that the antibody level in the five-year prognosis was mainly affected by the antibody level in the two-year prognosis (OR: 1.007, 95%CI: 1.005–1.009). In conclusion, the serum IgG antibodies of previous dengue fever cases can persist for a long time.

## 1. Introduction

Dengue fever, a mosquito-borne acute febrile viral disease caused by the dengue virus (DENV), has become one of the most severe infectious diseases with a steady rise in incidence globally [1]. Dengue is a major global public health issue affecting more than 100 countries or regions in the world, with 96 million of approximately 390 million infections developing symptomatic infections annually [2]. Following DENV infection, humans develop a mild to a severe clinical syndrome, with most patients presenting a flu-like self-limited disease, and a small number of patients developing severe dengue fever, such as dengue hemorrhagic fever (DHF) and dengue shock syndrome (DSS) [3].

In recent years, with the rise in the frequency of trade and tourism between China and other dengue-endemic countries, imported cases have increased significantly, thereby leading to more severe dengue outbreaks [4,5,6]. Guangzhou has always been considered the critical epicenter of dengue fever in Guangdong Province and even in the whole country [7]. In recent years, Guangzhou has been experiencing a high frequency of dengue fever transmission, especially in the 2014 record-breaking outbreak, accounting for nearly 80% of the total reported cases in mainland China [8]. It is believed that DENV is classified into 4 serotypes (DENV-1, DENV-2, DENV-3 and DENV-4). Studies have shown that dengue fever epidemics in Guangzhou in the past 10 years were still dominated by DENV-1 [9].

Dengue fever is considered to be one of the most public health problems in the world, mainly due to high incidence, wide epidemic and heavy disease burden. In fact, the current prevention of dengue fever is mainly limited to vector control, but effective vector control requires a detailed understanding of the drivers of DENV infection. There is no specific treatment for DENV infection, and the application of preventive vaccines is limited and being further evaluated. The ideal vaccine is one that can produce effective neutralizing antibodies against the four serotypes of DENV. However, a safe and effective vaccine has not been developed due to insufficient understanding of the pathogenesis of DSS and DHF at present. The Sanofi Pasteur vaccine, Dengvaxia, a recombinant chimeric live attenuated DENV vaccine based on a yellow fever 17D vaccine backbone, was evaluated in two large multicenter phase III trials [10]. Compared with seropositive vaccinators, antibody-negative people are at higher risk of severe dengue fever [11]. Therefore, it is crucial to understand the duration of antibodies in previous cases of dengue fever to develop vaccination strategies. Accordingly, it is of great significance to understand the dynamic changes of DENV antibody for the prevention of dengue fever.

Studies have shown that the IgM antibody in serum appears in the early stage of infection. It can be detected 3–5 days after the onset of the disease, persisting for 2–3 months [1]. It is believed that DENV IgG antibody usually appears 8–10 days after primary infection [1] and therefore can be used as long-term detection marker [12]. Previous studies have been committed to the measurement of specific antibodies in the acute phase and the recovery phase [13,14,15,16,17,18], and there is a lack of follow-up studies on the dynamic changes of IgG antibodies in patients. Luo [19] conducted a serological survey in symptomatic and asymptomatic individuals three years after infection in Zhejiang Province, China. They found that the positive rate of serum IgG antibody in dengue symptomatic individuals was still as high as 96.61% three years after infection and was only related to infection status. Furthermore, Tun detected IgG antibodies in an 84-year-old Japanese male who experienced acute febrile infection 70 years ago [20]. Cropp also found that the DENV IgG antibodies could exist in those who were infected more than 60 years earlier in Pacific and Southeast Asia [21]. It is believed that the secondary infection of DENV will increase the risk of DHF or DSS due to antibody-dependent enhancement (ADE), which means that the risk of severe dengue fever in these antibody-positive individuals will increase once they develop secondary heterotypic infection. According to Katzelnick’s research on the relationship between dengue neutralizing antibody titers and symptomatic infections, it was found that those with high neutralizing antibody titers were less likely to develop symptomatic infections [22]. However, the duration of DENV IgG antibodies and antibody concentration after DENV infection remain unclear in China.

In order to clarify the duration of specific antibodies in patients, we conducted a longitudinal serological survey on the antibody levels of previous infection cases in Guangzhou. We aim to provide a reference for the further development of prevention and control strategies of dengue fever and therapeutic antibodies. According to reports, there were 1249 local cases of dengue fever in Guangzhou in 2013, of which Liwan District ranked first among all urban counties with 781 cases [23]. In 2014, there were 4467 local cases in the Liwan District, which accounted for 11.96% (4467/37340) of all cases in Guangzhou [24]. Therefore, the study subjects were enrolled from Liwan District, the most important epicenter of DENV infection in Guangzhou.

## 2. Materials and Methods

### 2.1. Study Area

The Liwan District, located in the west of Guangzhou, is an important commercial and cultural center in Guangzhou. It covers a total area of 62.40 km^2^ and has a permanent resident population of 1012,000 (data sourced from Guangzhou Statistics Bureau, 2019. http://tjj.gz.gov.cn/). It has a warm, rainy subtropical climate. Furthermore, the Liwan District is suitable for mosquito growth and reproduction throughout the year, while *Culex* and *Aedes albopictus* are the dominant mosquito species.

### 2.2. Study Design and Subjects Enrollment

We conducted a longitudinal serological investigation of a group of dengue fever cases infected in 2013 and 2014. Eligible study subjects were those previously diagnosed with DENV-1 who had lived in Liwan District for a long time. Those with incomplete personal data and information were excluded from the study.

The study subjects were selected by using a two-stage sampling method. In the first stage, nine communities were randomly selected from eligible communities in Liwan District, that is, those with a large number of cases. In the second stage, individuals were randomly selected from selected communities. Sample size was calculated based on a power of 90% and a type 1 error of 5%. According to preliminary survey results, the average reduction concentration was expected to be 14 relative units per milliliter (RU/mL) with a standard deviation of 32 RU/mL. The calculation of sample size required was based on paired design, and then the above value was substituted into the Power Analysis and Sample Size software (version 15.0, NCSS, USA) to calculate the minimum sample size of 57. We assumed an extra 20% of missing data, so the final sample size was calculated to be 68 subjects.

A total of 70 patients with DENV-1 confirmed by the laboratory two years after infection were the study subjects. We enrolled them into the study after obtaining informed consent and prospectively followed them up in the fifth year of their prognosis. All study subjects were required to use standard questionnaires for face-to-face interviews to collect information about patients’ demographic characteristics, the history of dengue fever previous infections, basic medical history and flavivirus vaccination status, and to collect 5 mL of blood during follow-up visits.

All study subjects enrolled in the study received written informed consent, including children under 18 years of age, whose consent was provided by parents or guardians. This study was approved by the Ethics Committee of Guangzhou Center for Disease Control and Prevention.

### 2.3. Laboratory Methods

A blood sample was obtained from each subject from each survey and tested for DENV IgG antibodies using the Enzyme-Linked Immunosorbent Assay (ELISA) (NO. EI 266b-9601G, Euroimmun, Germany). The results were reported according to the manufacturer’s instructions, i.e., >22 units as positive results, 16–22 units as equivocal and <16 units as negative results. The concentration and absorbance of three calibrated samples were taken as horizontal and vertical coordinates to draw the standard curve, then the absorbance measured in the sample to be tested was substituted into the standard curve to calculate the antibody concentration of the sample.

### 2.4. Data Analysis

All questionnaire data were entered in duplicate using EpiData software (version 3.1, EpiData Association, Odense, Denmark). We used SPSS statistical software (version 25, Chicago, IL, USA) for statistical analysis. Univariate and multivariate analyses were performed using generalized estimation equations (GEE). The DENV IgG antibody titer of five years of prognosis was used as the response variable. The fixed effects included gender, age group, basic medical history (including hypertension, diabetes and viral hepatitis), vaccination of flavivirus (including Japanese encephalitis vaccine and the yellow fever vaccine), and the IgG antibody concentration of two years of prognosis. The variables with statistical significance in univariate analysis were included in the multivariate analysis. The statistical significance level was set at 0.05, and the odds ratio (OR) and 95% confidence interval (CI) were submitted.

## 3. Results

### 3.1. Study Population

A total of 70 DENV-1 patients with primary infection who had a prognosis of two years were selected, including 21 cases in 2013 and 49 cases in 2014. The demographic information of the study subjects shown in Table 1 illustrates that most participants were female (58.57%), aged < 60 years (82.86%) and educated in junior high school and below (34.29%). Overall, study subjects showed a diversity of occupation and annual household income, but most study subjects were business or service staff and most had an annual household income of 60,000–100,000 yuan.

### 3.2. Serum IgG antibody at 2 Follow-Up Visits

The comparison of IgG antibody positive rate and IgG antibody concentrations in the sample population was divided into subset variables by age or gender as shown in Table 2 and Table 3. The total serum IgG antibody positive rate was 100% (70/70) at two years post infection. However, the total serum IgG antibody positive rate was 95.71% (67/70) in those with a prognosis of five years. Furthermore, during the second follow-up, we found that the positive rate of serum IgG antibody in female (100%) was higher than that in male (89.66%) (*χ*^2^ = 4.431, *p* = 0.035), while the difference between different age groups was not statistically different (*χ*^2^ = 0.578, *p* = 0.447) (96.55% vs. 91.67%). (Table 2).

As shown in Table 3, the average concentration of total IgG antibody was (164.88 ± 31.34) RU/mL at two years of prognosis. It was found that there was no statistically significant difference in the concentration of IgG antibody between different gender and age groups (*p* > 0.05). Besides, the average concentration of total IgG antibody was (119.97 ± 47.20) RU/mL at five years of prognosis. According to statistics, the difference in serum IgG antibody concentration between different gender was statistically significant (*p* < 0.05), while there was no statistically significant difference between different age groups (*p* > 0.05).

As shown in Table 4, there was no statistically significant difference in the positive rate of total serum IgG antibody between two and five years of prognosis (*χ*^2^ = 3.066, *p* = 0.080). A normality test was performed on the difference in serum IgG antibody concentration between the two follow-ups, and the results showed that it was not normally distributed (W = 0.883, *p* = 0.04 < 0.05). We used the paired Wilcoxon signed rank-sum test and the results showed that the difference in serum total IgG antibody concentration between two follow-ups was statistically significant (Z = 7.154, *p* < 0.001).

### 3.3. Univariate Analysis

Table 5 shows the results of univariate analysis of factors influencing IgG antibody level at five years of prognosis. The serum antibody concentration at five years of prognosis was affected by the serum IgG antibody concentration at two years of prognosis (*p* < 0.05). Besides, compared with those without basic medical history, those with basic medical history had higher IgG antibody concentrations at five years of prognosis, that is, those with basic medical history had a slower rate of decline in serum antibody. However, the differences in other groups were not statistically significant (*p* > 0.05).

### 3.4. Multivariate Analysis

According to the results of the multivariable GEE analysis, the variable in the final equation was the serum IgG antibody concentration at the two-year prognosis and the odds ratio (OR) of the variable was 1.007 (95% CI: 1.005–1.009) (*p* < 0.05) (Table 6). In other words, those who had a high serum IgG antibody concentration at two years of prognosis had higher serum IgG antibody concentration at five years of prognosis, indicating that the high IgG antibody concentration declined more slowly over time.

## 4. Discussion

In this study, we selected study subjects who tested positive for DENV infection during the 2013–2014 DENV epidemic in Guangzhou. We then conducted longitudinal follow-up serological investigations in the second and fifth years of prognosis. It was found that the difference in serum IgG antibody positive rate between the two follow-ups was not statistically significant, indicating that the serum IgG antibody positive rate was relatively stable after DENV infection and would not decrease significantly over time. To a certain extent, it has very important biological significance, that is, IgG antibodies in the serum of dengue fever cases can be used as long-term detection markers.

Previous studies have found that serum antibodies can persist for years, or even decades, after human infection with flavivirus [25]. A study based on World War II veterans showed that the neutralizing antibody would last for 30–35 years after immunization with 17D yellow fever vaccine [26]. Furthermore, other studies in Japan have shown persistence of DENV IgG antibodies in serum after 60–70 years of DENV infection [20,21]. Undoubtedly, the results of our study strongly indicate that the primary infection of humans with DENV can produce longer-lasting immunity. It is worth noting that DENV secondary heterotypic infection will increase the risk of DHF or DSS due to ADE [27,28,29,30], that is, those people who are still IgG antibody-positive for many years after infection will have the possibility of severe dengue fever if secondary heterotypic infection occurs. Therefore, it is necessary to strengthen the monitoring of these antibody-positive people in order to take comprehensive measures to prevent the occurrence of severe dengue fever in the DENV secondary heterotypic infection.

It was further found that the serum antibodies maintain a stable detection rate in previous cases of dengue fever, while the antibody titer gradually decreases as time passed. Over time, the antibody concentration in serum samples of individuals with low IgG antibody concentration gradually decreased, and DENV IgG antibody could not be detected completely in some individuals after five years.

We found that the prognostic serum antibody concentration is only affected by the early antibody concentration. This may be due to the higher initial antibody concentration that has a slower rate of degradation over time. Katzelnick found that re-exposure to DENV, including homologous DENV serotypes, helps maintain high levels of neutralizing antibody titers [22]. Previous studies have shown that during DENV infection, while antibodies against the homozygous virus were produced, cross-protective antibodies could also be observed to provide short-term protection [31,32]. Therefore, the short follow-up period of the study resulted in the subjects being affected by the cross-protection effect, and secondary symptomatic infection could not be observed. A longer observation period is needed to further explore their relationship.

Our research has several limitations. First, we employed only anti-DENV IgG ELISA to determine the antibody-positive rate, and ELISA has the limitation of cross-reaction, especially with the Japanese encephalitis virus. However, there have been no reports of Japanese encephalitis in the Liwan District in the past 10 years, resulting in a low false-positive rate. Secondly, plaque reduction and neutralization experiments are needed in the future to further explore the titers of neutralizing antibodies and correct the false-positive results caused by the cross-reaction of flavivirus.

## 5. Conclusions

Despite the above limitations in our research, we have confirmed that the serum IgG antibody positive rate was relatively stable after DENV infection and would not decrease significantly over time. At the same time, we have a preliminary understanding of the relationship between prognostic serum antibody concentration and the factors affecting its attenuation. Neutralizing antibodies need further study by plaque reduction and neutralization experiments, as well as to correct the false positive results caused by the cross-reaction of flavivirus.

## Figures and Tables

**Table 1 ijerph-17-06580-t001:** Demographic information of the study subjects.

Factor		Number	Account (%)
Gender	Male	29	41.43
	Female	41	58.57
Age in years	<60 years	58	82.86
	≥60 years	12	17.14
Education	Junior high school and below	24	34.29
	High school/technical secondary school	23	32.86
	College and above	23	32.86
Occupation	Worker	11	15.71
	Professional technology/cadre/clerk	13	18.57
	Business/Service staff	15	21.43
	Other	31	44.29
Annual household income	Less than 30,000	14	20.00
	30,000–60,000	20	28.57
	60,000–100,000	26	37.14
	More than 100,000	10	14.29

**Table 2 ijerph-17-06580-t002:** Comparison of IgG antibody positive rate in different populations at each follow-up visit.

Variables	Total Number	First Follow-Up		*χ* ^2^	*p*	Second Follow-Up		*χ* ^2^	*p*
		Positive Number	Positive Rate (%)			Positive Number	Positive Rate (%)		
Age in years				NA	NA			0.578	0.447
<60	58	58	100.00			56	96.55		
≥60	12	12	100.00			11	91.67		
Gender				NA	NA			4.431	0.035 *
Male	29	29	100.00			26	89.66		
Female	41	41	100.00			41	100.00		

* Significance difference where *p*-value is less than 0.05.

**Table 3 ijerph-17-06580-t003:** Comparison of IgG antibody concentrations in different populations at each follow-up visit.

Variables	First Follow-Up		*t*	*p*	Second Follow-Up		*t*	*p*
	Average Value	Standard Deviation			Average Value	Standard Deviation		
Age in years			2.824	0.097			2.800	0.099
<60	161.23	31.69			118.62	44.64		
≥60	182.52	25.20			126.48	61.58		
Gender			0.373	0.543			11.603	0.001 *
Male	162.39	33.84			114.15	61.09		
Female	166.65	30.15			124.08	35.37		

* Significance difference where *p*-value is less than 0.05.

**Table 4 ijerph-17-06580-t004:** Comparison of total IgG antibody positive rate and concentration between two follow-up visits.

Follow-Up	Total IgG Antibody Positive Rate (%)	*χ* ^2^	*p*	Total IgG Antibody Concentration (Mean ± Standard Deviation)	*Z*	*p*
First	100.00	3.066	0.080	164.89 ± 31.56	7.154	<0.001 *
Second	95.71			119.97 ± 47.54		

* Significance difference where *p*-value is less than 0.05.

**Table 5 ijerph-17-06580-t005:** Analysis of univariate influencing factors of IgG antibody level at five years of prognosis.

Factor	*β*	*SE*	*χ* ^2^	*p*	OR (95%CI)
Age in years					
<60					1.0
≥60	−0.114	0.110	1.077	0.299	0.892 (0.719, 1.107)
Gender					
Male					1.0
Female	0.032	0.083	0.147	0.702	1.032 (0.877, 1.215)
Basic medical history					
No					1.0
Yes	0.218	0.076	8.126	0.004 *	1.243 (1.070, 1.444)
Vaccination					
No					1.0
Yes	0.146	0.127	1.321	0.250	1.158 (0.902, 1.485)
First follow-up antibody concentration	0.007	0.001	48.100	<0.001 *	1.008 (1.005, 1.010)

* Significance difference where *p*-value is less than 0.05. 95% confidence interval.

**Table 6 ijerph-17-06580-t006:** Analysis of univariate influencing factors of IgG antibody level at five years of prognosis.

Factor	*β*	*SE*	*χ* ^2^	*p*	OR (95%CI)
Basic medical history					
No					1.0
Yes	0.107	0.065	2.715	0.099	1.113 (0.980, 1.265)
First follow-up antibody concentration	0.007	0.001	40.089	<0.001 *	1.007 (1.005, 1.009)

* Significance difference where *p*-value is less than 0.05. 95% confidence interval.

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
