# Peer review of "Kinetics of IgG Antibodies in Previous Cases of Dengue Fever—A Longitudinal Serological Survey"

_ijerph, 2020, doi:10.3390/ijerph17186580_

Round 1
Reviewer 1 Report
To the Authors:
The study by Wu and colleagues aims to provide evidence of persistent IgG antibodies levels in patients from Guangzhou, China, using a longitudinal serological survey on antibody levels of previous infection cases. Other studies have reported evidence of IgG persistence; the novelty of this manuscript surrounds the findings in the Liwan District, thus the manuscript in its current form has a number of inconsistencies that should be addressed.
Major points:
Is there any evidence of multiple or sequential infections with different serotypes, given the persistence of IgG levels- does the ELISA test used detect other DENV serotypes? Is this linked to the high level of IgG reported; i.e other infections?
What were the clinical symptoms of the patients? What was the flavivirus vaccine given and at what stage since infection was it administered?
What is the significance of stating and acquiring the demographic characteristics, including education, occupation and income if not discussed or used as a parameter?
The authors do state this as a limitation but the ELISA methods should be improved to reliably differentiate between JE virus and other flavivirus infections from DENV infection; given the cross-reactivity of IgG.
Minor points:
The title is not completely clear. Suggest changing this to reflect the parameters of the study.
Be consistent with the way the study participants are referred to. They are currently referred to as patients/participants/subjects/research objects/study objects. The authors should use one term to describe these, so it is less confusing for the reader. Also be consistent in the dengue virus, dengue fever, DENV-1
Abstract:
Line 17 – Guangzhou City is referred to as just “Guangzhou” in the rest of the text. Be consistent.
Line 19 - "Influencing factors" should be more specific
Line 20 – “of antibodies” should be more specific.
Line 20 – DENV-1 abbreviation should be written as dengue virus type 1 (DENV-1) the first time
Line 20 – change “with 2 years postinfection” to “at 2 years post-infection”
Line 22 – change “dengue IgG” to “DENV IgG”
Line 24 – insert space after “equation”
Introduction:
The research question itself is not clearly stated. The study focusses on the detection and concentration of DENV IgG antibodies in patient serum over time, but there is little justification for researching this specifically. An indication of antibody changes over time in other virus infections, how these affect rates of reinfection and disease outcomes will greatly help to justify this study. It should be clearly explained why it is important to monitor antibody persistence and concentration over time.
Line 33 - introduce DENV abbreviation after “dengue virus”
Line 37 – Change “humans after dengue infection” to “Following DENV infection, humans with develop…”
Line 49 – “wide epidemiology” is confusing, consider re-wording
Line 51 – change “dengue” to “DENV”
Lines 53-55 - a brief discussion of the Sanofi Pasteur DENV vaccine, how it works and its limitations would add a lot of weight to the justification of this study
Line 59 – remove “in”
Line 59 – consider changing “lasting” to “persisting”
Line 60 – change “dengue” to “DENV”
Line 60 – give an indication on the timescale of the recovery phase. Is this on average a certain number of days post-infection, weeks, months or years?
Line 63 – change “antibody” to “antibodies” (specify IgG or IgM antibodies or both)
Line 64 – provide further details of the study by Luo and colleagues, reference 16.
Lines 61-66 – it is worth discussing the studies by Tun et al (2016) and Cropp et al (2007) that the authors discuss later on (refs 20 and 21) which examine antibody persistence at 60 and 70 years post-infection.
Line 66 - it is unclear what is meant by "related factors." An introduction to these related factors and how they may be necessary in providing immunity and how they may be measured would help guide the reader
Lines 68-71 – this is a long sentence which is potentially confusing. Consider splitting this into two
Line 71 – consider introducing the case numbers of Liwan District first before introducing the sampling and experimental model of this study.
Line 72 – change “dengue” to be more specific. Is it DENV infection or incidence of dengue fever?
Materials and Methods:
Line 78 – change “locating’ to “located”
Line 81 – remove “besides”
Line 87 – change “and they” to “who”
Line 87 – do the authors specifically mean DENV type 1 or dengue fever? If they are specifically referring to DENV-1 then this abbreviation needs to be introduced first. A brief statement in the introduction about the 4 different serotypes of DENV and their prevalence would work well.
Line 90 – the authors should specify what made a community eligible. It is previously mentioned what makes a participant eligible, but it is not clear what makes a whole community eligible
Line 93 – “the average reduction” is not clear. Please specify the reduction of what exactly
Line 93 – specify what RU is
Line 94 – please state why the minimum sample size is 57 (a reference for this needed?)
Line 108 - consider providing the product number as there are a few DENV ELISA kits from Euroimmun which detect both IgG and IgM from all 4 DENV serotypes.
Line 109 – change “following” to “according”
Line 109 – change “ie” to “;”
Lines 110-113 – “The concentration and absorbance of the three calibrated samples were taken as horizontal and vertical coordinates to draw the standard curve, and the antibody titers in the samples to be tested were calculated according to the standard curve.” This is a confusing sentence.
Line 111 – remove “the”
Line 115 – change “double-entered” to “entered in duplicate”
Results:
Lines 129-130 – “Of the total participants, 24 (34.29%) were attained junior high school and below, 23 (32.86%) were high school or technical secondary school, and 23 (32.86%) were college and above.” Consider re-wording.
Line 136 – “different populations” - consider changing this to state that the sample population was divided into subset variables by age or gender as shown in table 2 and table 3.
Line 139 – change “IgG antibody is” to “IgG antibody was”
Lines 143-150 - This is very confusing. It is important to be clear which result the authors are referring to in each sentence. While one states that there is no significant difference between age groups and gender, another sentence describes a significance between gender and age. Try to reinforce which follow-up is being referred to each time and also if referring to positivity or concentration. Also cite each table whenever a result is mentioned so that the reader can follow as easily as possible.
Line 143 – change “IgG antibody is” to “IgG antibody was”
Table 3 – this table could be more effectively shown as a graph although this is entirely down to author and journal preference
Table 3 – standard deviation missing from the second follow-up column. Need an explanation why if it is going to be omitted
Line 161 – p cannot be =0.000, must be p<0.001 or p<0.0001
Table 4 – change “follw-up” to “follow-up” in first column
Lines 164-177 - the results reported in sections 3.3 and 3.4 could be expanded further. and indication of what each result means will help guide the reader through your findings to the next section.
Line 173 – use GEE abbreviation
Line 175 - Again, expanding on what this means will make things much clearer.
Discussion:
Lines 179-180 – “we selected a group of cases confirmed by the laboratory during the dengue fever epidemic in Guangzhou from 2013 to 2014” – consider changing to two sentences e.g. “we selected subjects who tested positive for DENV during the 2013 DENV epidemic in Guangzhou. We then conducted...”
Line 188 – references needed on antibody persistence in flavivirus infections.
Line 189 – “can be continuously existed” needs re-wording
Line 190 – change “that IgG” to “persistence of DENV IgG”
Line 193 – first time use of ADE abbreviation. Need to expand and explain this concept
Line 199 – remove “However”
Lines 201-202 – “primary infection is related to viral load” reference needed for this as it does not currently appear to be a conclusion that can be made from the results presented in this study
Line 202 – “the IgG antibody titre was higher in patients with high viral load” citation needed
Lines 202-205 – “Over time, the antibody concentration in serum samples of individuals with low IgG antibody concentration gradually decreased, and even some individuals could not be detected dengue antibody completely after 5 years.” This sentence is confusing. There is little indication in the results section that low concentration IgG at 2 years leads to greater reduction in concentration at 5 years. If this is a conclusion that this study aims to make, then it should be expanded more in the results section so that this is clear to the reader.
Lines 206-207 – “We found that the prognostic serum antibody concentration is only affected by the early antibody concentration.” This should be more clearly indicated in the results section as it is not something that is clear.
Line 211 – the concept of “cross-protection” needs explaining
Lines 217-218 – How will plaque reduction and neutralizing experiments further expand on this research? More info is needed here.
Conclusions:
Line 223 – Again, it is not clear what is meant by related influencing factors. Be specific in concluding the findings of this work.
Line 224 – it is not clear how this study demonstrates the need to identify neutralizing antibodies. The authors should therefore expand on this.
Author Response
Dear editor:
We are sorry to revise this manuscript for too long. The revised version together with our responses to the reviewers have been uploaded. Please see the attachment.
We are looking forward to your response.
Yours sincerely
Xiujuan Wang

Reviewer 2 Report
The authors should expand the study to a larger number of patients with different characteristics of the disease, comorbidities or history of secondary infection in dengue.
Round 2
Reviewer 1 Report
Dear Authors,
I commend you on the efforts you have made to improve the manuscript.
An improved ELISA test should however be employed for future studies.
Kind Regards
This manuscript is a resubmission of an earlier submission. The following is a list of the peer review reports and author responses from that submission.